# Bioactive Compounds and Bioactivities of *Brassica oleracea* L. var. *Italica* Sprouts and Microgreens: An Updated Overview from a Nutraceutical Perspective

**DOI:** 10.3390/plants9080946

**Published:** 2020-07-27

**Authors:** Thanh Ninh Le, Chiu-Hsia Chiu, Pao-Chuan Hsieh

**Affiliations:** Department of Food Science, National Pingtung University of Science and Technology, Pingtung 91207, Taiwan; ltninh90@gmail.com (T.N.L.); cschiou@mail.npust.edu.tw (C.-H.C.)

**Keywords:** sprouts, microgreens, seedlings, broccoli, bioactive compounds, biological activities

## Abstract

Sprouts and microgreens, the edible seedlings of vegetables and herbs, have received increasing attention in recent years and are considered as functional foods or superfoods owing to their valuable health-promoting properties. In particular, the seedlings of broccoli (*Brassica oleracea* L. var. *Italica*) have been highly prized for their substantial amount of bioactive constituents, including glucosinolates, phenolic compounds, vitamins, and essential minerals. These secondary metabolites are positively associated with potential health benefits. Numerous in vitro and in vivo studies demonstrated that broccoli seedlings possess various biological properties, including antioxidant, anticancer, anticancer, antimicrobial, anti-inflammatory, anti-obesity and antidiabetic activities. The present review summarizes the updated knowledge about bioactive compounds and bioactivities of these broccoli products and discusses the relevant mechanisms of action. This review will serve as a potential reference for food selections of consumers and applications in functional food and nutraceutical industries.

## 1. Introduction

Over the past twenty years, the heavy dependence of human nutrition on the sustainability of agricultural production has been highlighted owing to rapid population growth and environmental degradation [1]. Increasing food and agricultural productivity has simultaneously imposed external expenses and consequences upon the financial resource, ecological system, and human health by excessive water use, soil occupation, fertilizers, pesticides, herbicides, and food waste treatment [2]. Thus, to meet the United Nations’ sustainable development goals, the agriculture and food sectors have been confronted with a major challenge: to provide adequate nutrition for global food demand while minimizing the negative impacts on the environment [3]. Furthermore, in recent years, improving public awareness about the healthy lifestyle framework has prompted the search for novel food sources, which are rich in essential nutrients and have positive effects on human health [4]. As a result, functional foods and nutraceuticals are gaining significant attention since these foods could provide both health benefits to reduce the risk of chronic diseases and basic nutrition [5,6]. Besides, many scientific projects and research groups are focusing on the recovery of food wastes and upgrading them into high-value byproducts to serve as functional ingredients in new product development [7,8,9,10]. Particularly, the issues of the food systems, including food safety, food security, and food sustainability, should be significantly addressed in the era of the coronavirus (COVID-19) pandemic crisis. The availability of bioactive constituent of food and functional foods may become essential as the demand of consumers to protect their immune system by applying healthier diets increases. The food industry should have innovations fast enough in the era of the COVID-19 pandemic crisis to offer functional foods fortified with bioactive compounds that promote health and support consumers’ immune system [11]. Thus, emerging technologies have been considerably studied and developed for the production of functional compounds from food sources, comprising extraction, separation, isolation, identification, and quantification [12,13]. Sprouts and microgreens are a new approach for functional foods with various advantages from the sustainability perspective, inclusive of eliminating the use of herbicides and pesticides, reducing the generation of food waste, and reaching a 10-fold increase of health-promoting phytochemicals compared to commercial adult plants [2,14].

Various types of young seedlings of vegetables and herbs, including sprouts, microgreens, and baby greens, are becoming popular due to their production technique and nutritional value [15]. Sprouts are distinct from microgreens, even though both products are consumed in an immature stage [16]. Sprouts constitute shoots and rootlets, obtained from germinated seeds that grow for 2–7 days, and are harvested when the cotyledons are still under developed and the true leaves have not yet emerged. Different to sprouts, microgreens are typically recognized by the full expansion of the cotyledon leaves and the appearance of the first true leaves that generally occurs within 7–21 days after sowing [16]. Sprouts and microgreens are ideally appropriate for indoor production with significant improvement of the safety properties, which obviates the presence of external contaminants, such as herbicides, pesticides, or heavy metals [17]. Additionally, seedling cultivation is a flexible and qualified process without the unwanted impacts of seasonal, climatic, and geographical variations [17].

Although small in size, sprouts and microgreens are the new class of specialty crops providing surprisingly intense flavors, vivid colors, and rich phytonutrient content. In addition to the normal nutritional value, the seedlings are considered as functional foods with substantial health-promoting properties [16]. Recent reports demonstrated that sprouts and microgreens contain higher levels of functional components such as phenolics, flavonoids, pigments, vitamins, and minerals than those of the mature counterparts of the same type [15,18]. Moreover, these secondary metabolites are positively associated with potential health benefits to strengthen the immune system and reduce the risk of developing several diseases, such as cardiovascular diseases, obesity, diabetes mellitus, and cancers [18]. Sprouts and microgreens are produced from the seeds of different vegetables, herbs, grains, or even some wild species. Candidate genotypes are expanding based on standards of taste and health, in which, the most utilized species are from the Cruciferae, Cucurbitaceae, Asteraceae, Chenopodiaceae, Lamiaceae, and Apiaceae families [17].

The Brassicaceae family, consisting of plants with an acrid taste (commonly known as Cruciferae), in general, and broccoli (*Brassica oleracea* L. var. *Italica*), in particular, has received considerable interest for the production of sprouts and microgreens [19]. The seedlings of broccoli (Figure 1) are an excellent source of health-promoting phytochemicals, including nitrogen-sulfur derivatives (glucosinolates and isothiocyanates), polyphenols (chlorogenic and sinapic acid derivatives, and flavonoids), minerals (selenium, potassium, and manganese), and vitamins (A, C, K, and B6), with concentrations much higher than the inflorescences [14,20,21,22,23]. Available data revealed that the consumption of broccoli sprouts and microgreens in a dietary serving plays an important role in human health and reduces the risk of chronic diseases. Numerous in vitro and in vivo studies indicated the chemical composition and multiple biological capacities of broccoli sprouts and microgreens, comprising antioxidant, anticarcinogenic, antimicrobial, anti-inflammatory, and antidiabetic activities [24,25,26,27]. The anticancer and antioxidant properties in particular have been comprehensively studied over recent years. Furthermore, human-based investigations have presented promising results about broccoli seedlings’ potential as a protective agent for several forms of cancer and other diseases [28,29,30].

To date, several reviews centering on the general health benefits of broccoli and its related compounds have been published [31,32,33]. However, these reports focused on broccoli florets only, whereas there are no detailed reviews available about the biologically functional properties of broccoli sprouts and microgreens. Therefore, the present review aimed to provide, for the first time, a detailed overview of broccoli sprouts and microgreens and an in-depth insight on their bioactive compounds and bioactivities from a nutraceutical perspective, with discussions on the relevant mechanisms of actions. This work was carried out by referring to the high-quality database of Web of Science, Scopus, PubMed, and Google Scholar to obtain updated articles. These search engines were employed for the following keywords alone and in combination: “*Brassica oleracea* L. var. *Italica*”, “broccoli”, “sprouts”, “microgreens”, “bioactive compounds”, “biological activities”, “antioxidant”, “anticancer”, “antimicrobial”, “health benefits”, and so on. In the present work, over ninety studies published from 2006 to 2020 about broccoli sprouts and microgreens were reviewed.

## 2. Bioactive Compounds

Bioactive compounds are extra-nutritional constituents found in foods, mainly in fruits, vegetables, and grains, which are capable of modulating metabolic processes and providing health-promoting benefits [34]. Regular consumption of broccoli seedlings could stimulate the natural defense systems and decrease the risk of chronic diseases, owing to their concentration of bioactive compounds several times higher than those of the mature florets [21]. Thus, they are considered a novel plant-derived functional food. Previous studies have extensively indicated that broccoli sprouts and microgreens contain a remarkably high amount of glucosinolates, phenolic compounds, and essential nutrients (Figure 2 and Tables 1–3). There are numerous methods employed for the extraction of bioactive compounds from herbs, vegetables, and fruits, including emerging technologies such as pressurized hot water extraction, microwave-assisted extraction, or pulsed electric extraction [35,36,37]. The separation, identification, and characterization of these compounds of broccoli seedlings, in particular, and the genus *Brassica*, in general, were investigated by both conventional and non-conventional technologies [38,39]. Among these technologies, the broccoli samples were mainly conducted by high-performance liquid chromatography (HPLC) coupled to diode array (DAD), ultraviolet-visible (UV–vis), electrospray ionization (ESI), or mass spectrometry (MS) detectors with C_18_ analytical columns. Gas chromatography combined with mass spectrometry (GC–MS) was also applied to characterize isothiocyanates, the hydrolysis products of glucosinolates. Besides, the spectrophotometric (UV–vis) detection was employed in most of the research to determine the total phenolic and flavonoid contents as the simplest procedure, if it was not required to determine a specific compound (Tables 1–3). It could be summarized that glucosinolates and related compounds are the major group of phytochemicals investigated in broccoli sprouts and microgreens, although phenolic compounds have also been analyzed in many studies (Figure 2).

### 2.1. Glucosinolates and Related Compounds

Glucosinolates (GLSs), nitrogen−sulfur compounds (β-_D_-thioglucoside-n-hydroxysulfates), are important plant secondary metabolites, almost exclusively found in the genus *Brassica*, specific to kale, cabbage, and broccoli [40]. GLSs are classified as aliphatic (derived from methionine, isoleucine, leucine or valine), aromatic (derived from phenylalanine or tyrosine), or indole (derived from tryptophan) groups. Aliphatic group is the major group of GLSs in almost all cruciferous seeds and seedlings of *B. oleraceae*, *B. napus*, *B. rapa*, and *R. sativus* [19]. In broccoli sprouts and microgreens, 26 compounds of GLSs were identified (Table 1). Among these compounds, the most abundant GLSs are glucoraphanin, glucoiberin, glucoerucin, glucobrassicin, and neoglucobrassicin [41]. In particular, glucoraphanin accounted for over 50% of the total GLSs content, which was quantified in the range of 605 to 1172 mg per 100 g of fresh weight (FW) [42,43,44].

Notably, GLSs are not the functional components in cruciferous vegetables, rather their hydrolysis products are the putative bioactive compounds [56]. When plant tissues are mechanically damaged, GLSs could be hydrolyzed by the enzyme myrosinase into a variety of degradation products, including isothiocyanates (ITCs), thiocyanates, nitriles, epithionitriles, and oxazolidines. In particular, ITCs have been proven to present strong anticarcinogenic activities [42]. As mentioned above, predominant GLSs in broccoli seedlings are glucoraphanin, glucoerucin, and glucobrassicin, which are enzymatically converted into sulforaphane (SFN), erucin (ERN), and iberin, respectively [56]. SFN is a naturally occurring inducer of phase II enzymes in human and animal bodies to detoxify cancer-causing chemicals. Thus, it can reduce the risk of different cancers, especially those of the bladder, colon, and lung [59]. To date, SFN was analyzed as a major compound among the total of 17 ITCs identified from broccoli sprouts and microgreens (Table 2). The total ITC content of broccoli seedlings was reported to be around 11 mg per 100 g of FW. Additionally, such seedlings were said to contain 90% SFN [19,66].

### 2.2. Phenolic Compounds

Besides GLSs and ITCs, another important group of bioactive constituents present in cruciferous vegetables is the phenolic compounds. They are secondary metabolites produced in plants through the phenylpropanoid and shikimate pathways [74]. Based on their structure, which comprises one or more aromatic rings with attached hydroxyl substituents, phenolic compounds can be categorized into various subgroups, such as phenolic acids, flavonoids, tannins, coumarins, lignans, quinones, stilbenes, and curcuminoids. These compounds have been mainly reported for antioxidant activity. Moreover, they are also associated with other health-promoting effects such as anticarcinogenic, antimicrobial, anti-inflammatory, and anti-aging properties [75]. Hydroxycinnamic acids and flavonoid glycosides are among the main phenolic compounds found in broccoli seedlings [21,76]. Using quantitative and qualitative analysis, 37 phenolic compounds were characterized in broccoli sprouts and microgreens, including 29 hydroxycinnamic acids and derivatives, and 8 flavonoids and derivatives. Their phenolic profile composed mostly of sinapic acid, gallic acid, flavonoids (quercetin and kaempferol), and other hydroxycinnamic acids (chlorogenic, caffeic acid, and ferulic acids) (Table 3). In general, total phenolic and flavonoid contents were determined to be in the ranges of 74–453 mg per 100 g of FW and 95–105 mg per 100 g of FW, respectively [43,65,77].

### 2.3. Other Compounds

Similar to other sprouted seeds, broccoli sprouts and microgreens are known for their high concentration of essential nutrient composition [17]. Essential nutrients are compounds that must be supplied from foods, such as vitamins, minerals, fatty acids, and amino acids. They are required for normal body functions, including DNA synthesis, energy production, and biosynthetic pathways [84]. Nonetheless, the number of studies, which determined the nutritional composition of broccoli seedlings and the variations occurring during germination, is significantly smaller than that of glucosinolates and phenolic compounds. These studies indicated that broccoli sprouts are expressly rich in vitamins (A, C, K, and folic acid), minerals (potassium, calcium, magnesium, and selenium), pigments (carotenoids and chlorophylls) and some other important nutrients (amino acids, fatty acids, and dietary fiber) [20,21,22,51,57,85,86]. In general, the biochemical composition of broccoli sprouts and microgreens depends mainly on the germination time, and 8-day-old sprouts displayed the highest nutrient concentration [22].

Among these compounds, carotenoids and chlorophylls were determined in several studies of broccoli sprouts [20,50,51,73,79]. Carotenoids are a group of isoprenoid molecules synthesized as secondary metabolites by all photosynthetic plants, including broccoli [87]. They have lipophilic antioxidant and immunomodulatory activities owing to the conjugated double bonds of the long polyene chain, which are capable of inhibiting reactive oxygen species and reducing oxidative damage [21]. Thus, carotenoids might prevent degenerative diseases, such as cardiovascular diseases, skin damage, diabetes, and several types of cancer [21]. The major carotenoids found in broccoli are β-carotene, α-xanthophyll (lutein), and β-xanthophylls (zeaxanthin, violaxanthin, and neoxanthin) [21]. In particular, β-carotene is the most typically studied carotenoids in broccoli sprouts and microgreens, due to its importance in medical science [50]. It supplies the human diet a substantial amount of vitamin A that is essential for organogenesis, tissue differentiation, immune function, and vision [87]. The total carotenoid (β-carotene) concentration of broccoli sprouts was reported in several studies with a range of 118 to 221 mg per 100 g of dry weight (DW) [20,21]. Chlorophylls are another group of light-absorbing pigments regularly present in broccoli sprout extract. They are primary metabolites that have a porphyrin structure [21]. Similar to carotenoids, they also have antioxidant, anti-mutagenic, and anti-inflammatory potential [21]. The content of chlorophyll in broccoli sprouts was commonly determined along with total carotenoid content in previous studies, ranging from 738 to 850 mg per 100 g DW [21,51]. Additionally, a study showed individual concentrations of carotenoids (lutein and neoxanthin) and chlorophyll (chlorophyll *a* and *b*) in broccoli sprouts [21].

### 2.4. Bioavailability of Bioactive Compounds

The human consumption of foods or nutraceuticals is not limited to bioactive compounds. In the human body, bioavailability is determined as the fraction of the dose administered that reaches the circulatory system and then distributes into target tissues so that the bioactive compounds are biologically available for exerting health-promoting benefits [88]. Many factors could affect the bioavailability of the dietary phytochemicals, including plant cell wall compositions, chemical structures, processing conditions, environmental stress, as well as an individual’s gastrointestinal system [56].

The effectiveness of a high consumption of broccoli sprouts in reducing the risk of cancer has been attributed mainly to their high content of GLSs. The daily intake of GLSs is difficult to estimate owing to the high variability of the constituent in cruciferous vegetables, particularly broccoli. A report for glucosinolate intake was indicated for European people ranging from 4.7–65 mg/day [32]. GLSs are broken down by both the plant enzyme myrosinase in the small intestine and the microbiota in the gastrointestinal tract into ITCs, which are reported to have the prominent anticancer effect [89]. After intake 2–3 h, the ITCs are absorbed from the small bowel and colon and metabolized by the mercapturic acid pathway. The metabolites are detectable in human urine and blood [53]. In broccoli sprouts, two of the most abundant GLSs are glucoraphanin and glucoerucin, and myrosinase hydrolysis of these GLSs form SFN and ERN, respectively. SFN, distinguished ITCs in broccoli sprouts, is one of the most potent naturally occurring inducers of phase 2 detoxification enzymes, boost antioxidant status, and protect animals against chemically induced cancer [40,90]. Its modes of action are involved with the repressor protein Kelch-like ECH associated protein 1 (Keap1), nuclear factor erythroid 2 p45-related factor 2 (Nrf2), and genes, which contain an antioxidant responsive element (ARE) [62]. Thus, it is important for the inclusion of potential cancer chemopreventive agents in dairy foods. Moreover, it has shown an inhibitory effect for urease from *Helicobacter pylori*, a human pathogen [40]. The formation of other breakdown products from GLSs by intestinal microbiota is not well documented. For example, *Bifidobacterium* strains belonging to the human intestinal microbiota can in vitro metabolize the GLSs to nitriles. It is also known that several microorganisms are able to convert nitriles into ammonia and organic acids [89].

Phenolic compounds are substantial micronutrients in the human diet and the health benefits of polyphenols depend on the amount consumed and on their bioavailability [91]. The dietary intake of phenolic compounds has been associated with health-promoting effects, such as antioxidant, anti-aging, antiproliferative, and anti-inflammatory activities [19]. It has been reported that the average dietary intake of phenolic compounds for humans is about 780–1058 mg/day, including 50% of hydroxycinnamic acids, 20–25% of flavonoids, and 1% of anthocyanins [87]. Bioavailability differs significantly among phenolic compounds so that the most abundant compounds in the human diet are not necessarily those leading to the highest concentrations of active metabolites in target tissues [91]. There are two directions for the digestion of dietary phenolics, comprising digestion along the gastrointestinal tract and digestion inside the enterocytes. These digestions are based on hydrolase enzymes, which are available in the intestinal lumen, brush border, and enterocyte [88]. Many studies showed that polyphenols are absorbed in both the small intestine and the large intestine by intestinal enzymes after microbial digestions [88]. In the human small intestine and stomach, gallic acid and caffeic acid are the most well-absorbed compounds (95%), followed by catechins (20%). The least well-absorbed polyphenols are proanthocyanidins and the anthocyanins. They are pH-sensitive, which could be broken down in the stomach and readily absorbable. Flavonoid group is one of the group molecules with molecular weights > 500 Da so that it has a low bioavailability level (1%) because molecules are improbable to be transported through passive diffusion pathways [76,88,91].

The dietary intake of carotenoids, particularly vitamin A, has been linked to the protection of DNA, proteins, and lipids from oxidative damage [87]. Digested carotenoids decrease oxidative DNA damage so that they could protect colon cells against the stress of reactive oxygen species [87]. The dietary chlorophylls from fresh fruits and vegetables, which compose of chlorophyll *a* and *b*, showed distinct biological potentials, including wound healing, the control of calcium oxalate crystals, the modulation of xenobiotic metabolism, and the induction of apoptosis [21].

## 3. Biological Activities

Broccoli seedlings and their bioactive components exhibit many potential health-promoting roles. In the last decade, the beneficial effects, including antioxidant, anticarcinogenic, antimicrobial, anti-inflammatory, and several other properties, have been widely investigated in a number of in vitro and in vivo studies and topically applied in clinical trials [24,25,92]. Particularly, these studies mostly focused on the antioxidant and anticancer activities of broccoli sprouts and microgreens owing to the functions of glucosinolates and phenolic compounds (Table 4 and Table 5). In this section, the findings of different biological activities of broccoli sprouts and microgreens will be discussed with the possible mechanisms of action (Figure 3).

### 3.1. Antioxidant Activity

The overproduction or incorporation of free radicals, such as reactive oxygen species (ROS), cause oxidative damage to biomolecules and consequently lead to many chronic diseases, including neurodegenerative diseases, cardiovascular diseases, and certain age-related cancers [93]. ROS are previously thought to form in mammalian cells almost exclusively as a consequence of mitochondrial metabolism. Recently, it has been demonstrated that cellular enzymes known as nicotinamide adenine dinucleotide phosphate (NADPH) oxidases produce a considerable amount of ROS in the human body. Moreover, other cellular sources of ROS involve neutrophils, monocytes, endothelial cells, xanthine oxidases, cytochrome P450, lipoxygenases, and nitric oxide syntheses [94]. Hence, ROS has distinct effects on normal physiological processes, oxidative stress/regulation, metabolic diseases, and chronic inflammation. Targeting ROS is involved in antioxidant, anti-inflammatory, antidiabetic, and anti-obesity therapeutics [95]. Antioxidants are divided into natural and synthetic compounds, which could remove free radicals, inhibit ROS formation, and scavenge ROS [93]. Previous studies on broccoli sprouts and microgreens have demonstrated that they are recognized for their variety of naturally occurring antioxidants, comprising both nutritional antioxidants like vitamins (especially, ascorbic acid and α-tocopherol), and non-nutritional antioxidants such as carotenoids and phenolic compounds [96]. It was reported that these compounds are responsible for 80−95% of the total antioxidant capacity in broccoli sprouts [14]. The antioxidant activity of broccoli sprouts and microgreens has been determined in numerous studies using various methods. The capacity of broccoli sprouts and microgreens for chelating Fe2^+^ and scavenging free radicals such as DPPH (2,2-diphenyl-1-picrylhydrazyl) and ABTS (2,2′-azino-bis-3-ethylbenzothiazoline-6-sulphonic acid) was demonstrated in numerous studies (Table 4).

Among the possible methods, DPPH radical scavenging assay, ABTS radical cation decolonization assay, and ferric reducing antioxidant power (FRAP) assay were commonly employed in previous studies **(Table 4**). The DPPH method is speedy, simple, and low cost in comparison to other test models. DPPH, a dark crystalline molecule, is a stable chromogen radical formed by the delocalization of the spare electron over the molecule. The DPPH scavenging assay is based on the electron donation of antioxidants to neutralize DPPH radical. The reaction occurs with the loss of the violet color of DPPH that is measured at 517 nm, and the discoloration acts as an indicator of the antioxidant effectiveness [31]. The ABTS decolonization assay is appropriate for both hydrophilic and lipophilic antioxidants. It uses a spectrophotometer to measure the ability of antioxidants to scavenge the stable radical cation ABTS^+^, a blue-green chromophore molecule with absorption at 734 nm. Antioxidants can neutralize and decolorize the radical cation ABTS^+^ by electron or hydrogen atom donations [97]. The FRAP assay was originally employed to measure reducing power in plasma, but it has been expanded for other biological fluids and plant extracts. Thus, it is applicable for both in vitro and in vivo experiments. The FRAP mechanism is based on electron transfer rather than hydrogen atom transfer. The assay demonstrates the ability of antioxidants to reduce ferric iron. It measures the reduction of the ferric ion (Fe^3+^)–ligand complex to the blue-colored ferrous (Fe^2+^) form at low pH by antioxidants (absorption at 593 nm) [98].

Moreover, another mechanism of antioxidant activity of broccoli sprouts and microgreens have been confirmed in several in vitro and in vivo examinations by inhibiting the activity of prooxidant enzymes such as lipoxygenase (LOX), and xanthine oxidase (XO), and activating antioxidant enzymes such as peroxidase (POD), catalase (CAT), superoxide dismutase (SOD), glutathione peroxidase (GPX), NAD(P)H-quinone acceptor oxidoreductase 1 (NQO1), and heme oxygenase-1 (HO-1) (Table 4). Prooxidant enzymes are considered as the main biological source of superoxide radicals, whereas antioxidant enzymes could eliminate the excess of reactive oxygen species and reduce oxidative damage during senescence [48,83]. For example, enzyme activities of cytosolic glutathione peroxidase (GPX1), thioredoxin reductase(TR) in the thyroid, plasma glutathione peroxidase (GPX3), and ferric reducing ability of plasma (FRAP) significantly increased in response to broccoli sprouts ingestion in 4-week-old Wistar rats [72].

In summary, previous studies showed that broccoli sprout extracts rich in vitamins, carotenoids, and phenolic compounds showed very high antioxidant activity in both in vitro and in vivo tests (Table 4). Thus, they could be applied for antioxidant therapeutics to reduce the risk of chronic diseases caused by ROS. Interestingly, although the antioxidant capacity of broccoli seedlings has been reported comprehensively, it is still attracting considerable attention from investigators and researchers at present [20,24,80].

**Table 4 plants-09-00946-t004:** Antioxidant activity of broccoli sprouts and microgreens.

Study Type	Germination Time	Antioxidant Activity Assays	Sample Treatment	Ref.
In vitro	1–9 days	DPPH	50% methanol	[80]
In vitro	3–14 days	DPPH, ABTS	70% methanol	[54]
In vitro	5 days	DPPH, ABTS, FRAP	70% methanol	[78]
In vitro	9 days	DPPH, FRAP	90% methanol	[51]
In vitro	4–12 days	DPPH, FRAP	Methanol	[14]
In vitro	3–9 days	DPPH, FRAP, POD, CAT, SOD, GPX	75% methanol	[48]
In vitro	7 days	TEAC, ORAC	Methanol	[20]
In vitro	2–9 days	T-AOC kit	50% ethanol	[69]
In vitro	9 days	DPPH, T-AOC Kit	75% methanol	[44]
In vitro	4 days	T-AOC Kit	50% methanol	[63]
In vitro	5–7 days	DPPH, ABTS, FRAP	Methanol	[79]
In vitro	7 days	FRAP	50% ethanol	[57]
In vitro	7 days	FRAP	Boiling water	[58]
In vitro	12 days	DPPH, ABTS	Ethanol	[45]
In vitro	3–7days	DPPH	Dichloromethane	[67]
In vitro	6 days	ABTS, FRAP, CHEL, LPO, LOXI, XOI, CAT	50% ethanol	[76]
In vitro	3–10 days	DPPH, ABTS, FRAP	70% ethanol, 70% methanol, boiling water	[24]
In vitro	3 days	DPPH	80% methanol	[65]
In vitro	5–12 days	DPPH, CHEL, LPO	Boiling water	[81]
In vitro	3–11 days	DPPH	80% methanol	[22]
In vitro	5–9 days	DPPH	70% ethanol	[77]
In vitro	NS	TEAC	Buffer solution	[46]
In vitro	7 days	DPPH, MA/GC, Carboxylic Acid	Hexane, dichloromethane, acetone	[96]
In vitro	8 days	DPPH, ABTS, FRAP	60% methanol	[82]
In vitro	6 days	ABTS, FRAP, CHEL, LPO, LOXI, XOI, CAT, SOD	Buffer solution	[83]
In vitro	9 day	ORAC, TEAC	Buffer solution	[73]
In vitro	6 days	LOXI, XOI	50% methanol	[99]
In vivo	4 days	FRAP, GPX	Male Wistar rats’ plasma	[72]
In vivo	NS	NQO1, HO-1	Female SKH-1 mice’ skin tissue	[100]

ABTS, 2,2′-azino-bis-3-ethylbenzothiazoline-6-sulphonic acid decolonization activity; DPPH, 2,2-diphenyl-1-picrylhydrazy radical scavenging activity; FRAP, ferric reducing-antioxidant power; TEAC, trolox equivalent antioxidant capacity; ORAC, oxygen radical absorbance capacity; POD, peroxidase activity; CAT, catalase activity; SOD, superoxide dismutase activity; GPX, glutathione peroxidase activity; CHEL, metal chelating activity; LPO, inhibition of lipid peroxidation; LOXI, inhibition of lipoxygenase; XOI, inhibition of xanthine oxidase; NQO1, NAD(P)H-quinone acceptor oxidoreductase 1; HO-1, heme oxygenase-1; MA/GC, malonaldehyde/gas chromatography.

### 3.2. Anticancer Activity

In the last decade, many studies and projects have supported the protective effects of natural products in cancer prevention. The protective role against cancer has been mostly attributed to the high content of GLSs, typically found in cruciferous vegetables [40]. Broccoli sprouts and microgreens have been valued as a rich source of GLSs and their hydrolysis products (ICTs, particularly SFN), which are a well-known class of cancer chemotherapeutic agents that work by inducing apoptosis and arresting cell cycle progression (Table 1 and Table 2). The potential mechanisms of action mainly involve the inhibition of proliferation and the induction of apoptosis in cancer (Table 5). Hence, to demonstrate the anticancer activity of broccoli sprouts and microgreens, many in vitro tests have been carried out to determine the antiproliferative activity. Their results indicated that broccoli seedlings exert strong cytotoxicity against different types of cancer cell lines, including hepatocellular carcinoma cells (HepG2), prostate carcinoma cells (PC-3, AT-2, and SUM159), lung carcinoma cells (A549), and colorectal adenocarcinoma cells (Caco-2, and HT-29) (Table 5). In these studies, the highly effective IC_50_ values were found to be from 29 to 190 µg/mL. On the another hand, the selectivity of broccoli seedlings was reported on normal skin fibroblasts (BJ), normal colon fibroblasts (CCD18-Co), and normal liver cells (FL83B) by displaying no toxic effects on their viability after the treatment of broccoli samples (Table 5).

Cancer is essentially a disease of uncontrolled cell division. It is caused by an imbalance between cell proliferation and apoptosis [101]. Its development and progression are generally associated with the disorder of cell cycle regulators’ activity. Defects in the programmed cell death mechanism also make a key contribution to tumor pathogenesis [101]. Thus, most chemotherapeutic agents exert their cytotoxic activity against cancer cells, since they cause DNA damage and activate a complex signaling network resulting in cell cycle arrest and apoptosis induction [24]. Targeting the cell cycle phase and the checkpoint signaling pathway, which leads to the arrests at G1/S or G2/M phases, would provide a promising opportunity for cancer treatment. Besides, triggering cell apoptosis could be an effective strategy for potential chemotherapeutic agents [24,25,73,101]. To confirm the antiproliferative activity of broccoli sprouts and microgreens, several in vitro studies revealed the mechanism of cell death based on inducing cell cycle arrest and apoptosis (Table 5). Their results showed cell cycle arrests (obviously at G0/G1 and S phases) and significant increases in cell percentage with subG1 DNA content, which is considered as a marker of apoptosis. Besides, mitochondrial changes might activate the intrinsic apoptotic pathway. The loss of mitochondrial membrane potential (MMP) leads to the activation of several proteins linked to apoptosis, such as caspase-9 and cytochrome *c* [24]. Hence, a few studies indicated the notable decrease in MMP levels of cancer cells after treatment by broccoli seedlings to prove the mechanism of programmed cell death [24,73].

Furthermore, some in vivo carcinogenesis models were employed to demonstrate the anticancer effects and related molecular mechanisms of broccoli sprouts and microgreens, such as C57BL/6 mice, BALB/c mice, SKH-1 hairless mice, and Sprague-Dawley rats (Table 5). SFN extracted from broccoli sprouts showed the inhibition of breast cancer stem cells and downregulate the Wnt/β-catenin self-renewal pathway in a nonobese diabetic/severe combined immunodeficient xenograft model [102]. SFN isolated from broccoli seeds and sprouts also exhibited anticancer effects in lung cancer cell lines and nude BALB/c mice with lung cancer xenograft by inhibiting the PI3K-AKT signaling pathway [25]. The transgenic adenocarcinoma of the mouse prostate model with broccoli sprout intake demonstrated a significant decline in prostate cancer occurrence and HDAC3 protein expression in the epithelial cells of the prostate. HDAC3 is one of the histone deacetylase enzymes that turn off tumor suppressor genes and promote the expression of oncogenes [103]. The broccoli sprout diet administered to adult Her2/neu mice showed both preventive and suppressive effects on mammary cancer. These protective effects were associated with tumor- and epigenetic-related gene expression, and changed histone acetylation, DNA methylation, and DNA hydroxymethylation levels [104]. The dietary administration of broccoli sprout extract to the rat model inhibited bladder cancer development by inducing glutathione S-transferase and NAD(P)H-quinone oxidoreductase 1 in the bladder, which are important enzymes against oxidants and carcinogens [105].

**Table 5 plants-09-00946-t005:** Anticancer activity of broccoli sprouts and microgreens.

Study Type	Germination Time	Subjects	Potential Mechanisms	Ref.
In vitro	3–7 days	HepG2, CT26 cells	Inhibiting cell proliferation	[45]
In vitro	5 days	PC-3 cells	Inducing apoptosis; increasing ROS generation	[67]
In vitro	4 days	HepG2, SW480, BJ cells	Inhibiting cell proliferation; inducing apoptosis	[70]
In vitro	4–12 days	MAT-LyLu, AT-2 cells	Inhibiting cell proliferation and motility	[76]
In vitro	5 days	HepG2, Caco-2, A549, FL83B cells	Inducing cell cycle arrest and apoptosis; decreasing MMP level	[24]
In vitro	5 days	Caco-2, HT-29, HepG2 cells	Inhibiting cell proliferation	[59]
In vitro	8 days	U251, MCF-7, 786-0, NCI-H460, HT-29 cells	Inhibiting cell proliferation	[82]
In vitro	5 days	AGS cells	Inhibiting cell proliferation and motility	[83]
In vitro	7 days	LNCaP, PC-3, DU-145 cells	Decreasing PSA secretion; inducing apoptosis	[71]
In vitro	NS	MCF7, SUM159 cells	Inhibiting cell proliferation; inducing apoptosis	[102]
In vitro	NS	A549, H460, H446, HCC827, H1975, H1299 cells	Inhibiting cell proliferation; inducing apoptosis	[25]
In vitro	NS	Caco-2, CCD18-Co cells	Inducing cell cycle arrest and apoptosis; decreasing MMP level; increasing ROS generation	[73]
In vivo	NS	Female NOD/SCID mice	Eliminating breast CSCs in vivo; downregulating Wnt/β-catenin pathway	[102]
In vivo	NS	Female nude BALB/c mice	Inhibiting the PI3K-AKT signaling pathway	[25]
In vivo	3 days	Female SKH-1 hairless mice	Stabilizing p53; inducing phase 2 enzyme; inhibiting iNOS upregulation	[106]
In vivo	NS	Male TRAMP mice in C57BL/6 background	Decreasing HDAC3 protein expression	[103]
In vivo	NS	SV40 and Her2/neu mice	Modulating epigenetic pathways; regulating epigenetic-controlled gene expression	[107]
In vivo	NS	Female Her2/neu mice	Regulating tumor- and epigenetic-related gene expression; increasing tumor suppressor gene expression	[104]
In vivo	3 days	Female Sprague-Dawley rats	Inducing GST and NQO1	[105]

HepG2, hepatocellular carcinoma cell line; PC-3, AT-2, MCF-7, LNCaP, DU-145, and SUM159, prostate carcinoma cell lines; A549, NCI-H460, H460, H446, HCC827, H1975 and H1299 lung carcinoma cell lines; CT26, SW480, Caco-2, and HT-29, colorectal adenocarcinoma cell lines; U251, glioma carcinoma cell line; 786-0, kidney carcinoma cell line; AGS, gastric adenocarcinoma cell line; MAT-LyLu, malignant carcinoma cell line; BJ, normal skin fibroblast cell line; FL83B, normal liver cell line; CCD18-Co, normal colon fibroblast cell line; NOD/SCID, nonobese diabetic/severe combined immunodeficient mice; TRAMP, transgenic adenocarcinoma of the mouse prostate mice; SV40, transgenic C3(1)-SV40 Tag (FVB-Tg(C3-1-TAg)cJeg/JegJ) mice; Her2/neu, transgenic FVB/N-Tg(MMTVneu)202Mul mice; MMP, mitochondrial membrane potential; PSA, prostate-specific antigen; CSCs, cancer stem cells; HDAC, histone deacetylase enzymes; GST, S-transferase; NQO1, NAD(P)H-quinone acceptor oxidoreductase 1.

### 3.3. Antimicrobial Activity

The antimicrobial capacity of broccoli has been reported in several publications; however, most of these studies focused on broccoli florets. There are a few reports that displayed the detrimental effect of broccoli sprouts on pathogenic bacteria. A study investigated broccoli sprouts with high levels of gallic acid, esculetin, ferulic acid, and myricetin have the antibacterial activity against foodborne pathogens, including both Gram-positive bacteria (*Staphylococcus aureus* and *Bacillus subtilis*) and Gram-negative bacteria (*Salmonella typhimurium* and *Escherichia coli*), with minimum inhibition concentration (MIC) values from 390 to 1560 µg/mL [24]. This study revealed that the Gram-positive bacteria were more sensitive to broccoli sprout extract than the Gram-negative bacteria, similar to the previously published results of other plant extracts [108,109]. The possible reason might be the structural differences in the cell wall between these two classes of bacteria. Gram-negative bacteria are surrounded by an additional outer membrane, which is a hydrophilic layer to prevent the access of many substances including natural compounds [108,109]. Thus, broccoli sprout extract was demonstrated to be more active on Gram-positive bacteria.

Another in vitro study reported the powerful bactericidal activity of broccoli sprouts against *Helicobacter pylori* by the presence of sulforaphane as the major anti-*Helicobacter* active compound, with highly active inhibition zones (>5 cm) [26]. Many reports showed that plant-derived bioactive compounds, such as phenolics, flavonoids, and glucosinolates, can inhibit the growth and activity of various microorganisms [110]. With the diversity in the molecular structure and chemical composition, these compounds can perform distinct antimicrobial effects, such as destabilization of the plasma membrane or inhibition of extracellular enzymes [110]. The antimicrobial activity against pathogenic bacteria, yeast, and phytopathogenic fungi was also proven by the presence of many antimicrobial peptides in broccoli floret extract [111].

Moreover, the daily intake of sulforaphane-rich broccoli sprouts for 2 months was demonstrated to reduce gastric bacterial colonization and attenuate gastritis in *Helicobacter pylori*-infected mice and patients [23]. In another clinical trial, the high-sulforaphane broccoli sprouts powder also showed a considerable effect on *H. pylori* eradication in type 2 diabetic patients with positive *H. pylori* [112].

Collectively, these findings indicate that broccoli sprout extracts might serve as a potential source of antimicrobial agents in the food and pharmaceutical industry.

### 3.4. Anti-Inflammatory Activity

Several studies proved that broccoli sprouts and their bioactive components possess anti-inflammatory activity. Hence, the application of broccoli sprouts and microgreens has potential in the prevention and treatment of inflammatory bowel diseases, such as ulcerative colitis and Crohn’s disease [27]. The anti-inflammatory mechanisms of broccoli sprouts are probably associated with the nuclear factor-kappa B (NF-κB) and nuclear factor erythroid 2-related factor 2 (Nrf2) signaling pathways [27]. Both in vitro and in vivo experiments, as well as clinical trials, mainly exhibit anti-inflammatory effects of broccoli sprouts, which showed high concentrations of sulforaphane and a group of phenolic compounds, including anthocyanins, isoquercetin, chlorogenic and cinnamic acids, via inhibiting inflammatory mediators such as a nitric oxide (NO), decreasing the levels of proinflammatory cytokines such as tumor necrosis factor α (TNF-α), interleukin-6 (IL-6), and IL-1β, and increasing the levels of anti-inflammatory cytokines such as IL-10 and IL-22 [27,28,55,72,113,114]. For example, a study showed that sulforaphane-enriched broccoli sprouts inhibited activation of the NF-κB signaling pathway and the secretions of inflammatory proteins (inducible nitric oxide synthase (iNOS), cyclooxygenase 2 (COX-2), TNF-α, IL-6, IL-1β, and prostaglandin E2 (PGE2)) in microglial cells (BV2) and male ICR (Institute of Cancer Research) mice. In this study, broccoli samples also upregulated the expression of Nrf2 and heme oxygenase-1 (HO-1) in normal BV2 cells and against scopolamine-induced amnesia in mouse brain tissue samples [27]. To analyze immunological parameters in the Wistar rats with iodine deficiency, the significant decrease in IL-6 level, along with no significant differences in IL-10 concentration, were observed after adding broccoli sprouts to the diet [72]. High-sulforaphane broccoli sprouts also indicated favorable effects on inflammatory markers by reducing concentrations of serum high-sensitive C reactive protein (hs-CRP), IL-6, and TNF-α in type 2 diabetic patients [28].

### 3.5. Antidiabetic Activity

Broccoli is one of the few vegetables that is considered as a supplementary treatment for type 2 diabetes and for the prevention of its long-term complications [31]. A few studies investigated the potential benefits of broccoli sprouts and microgreens for patients with diabetes. A study determined the effects of broccoli sprout powder on insulin resistance in type 2 diabetic patients as a new approach by the use of its bioactive constituents. The results showed that broccoli sprout powder containing a high concentration of sulforaphane may significantly decrease in serum insulin concentration and lessen complications of diabetes [29]. In another report, the potential efficacy of sulforaphane extracted from young broccoli sprouts has been confirmed as an effective option for supplementary treatment in type 2 diabetes. It could induce some peroxisome proliferators-activated receptors, which contribute to glucose homeostasis in hyperglycemia and oxidative conditions [92].

### 3.6. Anti-Obesity Activity

Obesity has become a worldwide health problem and leads to adverse metabolic disorders, such as cardiovascular disease and type 2 diabetes. A couple of studies documented positive actions of broccoli sprouts on obesity by regulating lipid metabolism. In a double-blind clinical trial, broccoli sprout powder as supplementary treatment in type 2 diabetic patients could have beneficial effects on lipid profiles and oxidized low-density lipoprotein ratio (OX-LDL/LDL), as risk factors for obesity and cardiovascular disease [30]. Another mechanism to regulate lipid metabolism of broccoli sprouts may be associated with sulforaphane capacity to induce the Nrf2 pathway [92]. A recent study revealed that glucoraphanin extracted from broccoli sprouts can decrease the lipid accumulation and increase the Nrf2 activation. It leads to the downregulation of the expression of multiple genes involved in gluconeogenesis and lipogenesis, thereby alleviating obesity and diabetes [115].

### 3.7. Other Effects

Several other health-promoting effects of broccoli sprouts and microgreens have also been explored by in vitro, in vivo, and clinical research. A couple of studies indicated that broccoli sprouts are a good source of bioactive compounds that act as xanthine oxidase (XO) inhibitors [76,83,99]. XO is the enzyme that catalyzes the metabolism of hypoxanthine and xanthine into uric acid, so inhibiting XO activity may be potentially useful in the treatment of gout or other XO-induced diseases [99]. Broccoli sprout supplementation during pregnancy and the early newborn period could reduce brain injury following placental insufficiency in the newborn rats. The findings provide a new approach for the prevention of cerebral palsy and disabilities related to placental insufficiency [116]. The analgesic and antinociceptive effects of broccoli sprout extract were proven by the opioid mechanism using two in vivo experimental models of nociception. This study suggests the potential activity of broccoli sprouts in pain therapy [117]. In a double-blind study, the short-term ingestion of broccoli sprout homogenates showed the beneficial effects on nasal responses to the influenza virus in smokers by significantly decreasing virus-induced markers of inflammation and reducing the virus quantity. It may be a promising strategy for preventing influenza risk in smokers and other people exposed to airborne pollutants [118].

## 4. Conclusions and Future Perspectives

In recent years, broccoli sprouts and microgreens are one of the most consumed vegetable products. They have gained recognition as functional foods or nutraceutical foods by the increasing interest of consumers for diets that support health and longevity. Consequently, the health-promoting compounds of broccoli seedlings have been extracted and isolated to integrate into food and pharmaceutical product formulations. Over the past ten years, the extraction, isolation, and characterization of the functional properties of these broccoli products have been demonstrated by numerous scientific publications. In this review, we summarized for the first time the research findings of the last decade into bioactive constituents, bioactivities, and molecular mechanisms of broccoli sprouts and microgreens. They have been proven to present an abundant source of important natural compounds, including glucosinolates, phenolics, flavonoids, vitamins, minerals, and pigments. In particular, glucosinolates and their hydrolysis products are the main components analyzed, followed by phenolic compounds with a detailed profile. Moreover, the previous studies have focused on several biological activities of broccoli seedlings, such as antioxidant, anticancer, antimicrobial, and anti-inflammatory, as well as the potentially beneficial effects for patients with cancers, diabetes, and obesity. In general, they are non-toxic or have low toxicity.

Hopefully, this updated review, in addition to being a reference for consumers’ food selections, might attract more attention to broccoli sprouts and microgreens and their further applications in the food and nutraceutical industries, or even in clinical studies as cancer chemopreventive agents.

In the future, more bioactive compounds in broccoli sprouts and microgreens will be isolated, identified, and evaluated. Further investigations are required for exploring unrecognized biological functions and their underlying mode-of-action and molecular mechanisms in both in vitro and in vivo models, such as cardiovascular protective, hepatoprotective, neuroprotective, or enzyme inhibitory potentials. Besides, well-designed clinical trials should be carried out to confirm these health-promoting benefits of broccoli seedlings on humans.

## Figures and Tables

**Figure 1 plants-09-00946-f001:**
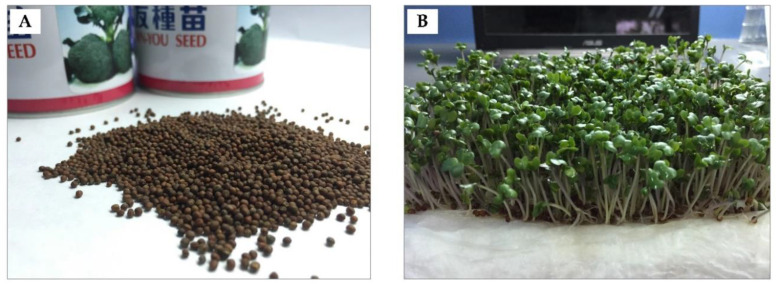
(**A**) Broccoli seeds. (**B**) Five-day-old broccoli sprouts.

**Figure 2 plants-09-00946-f002:**
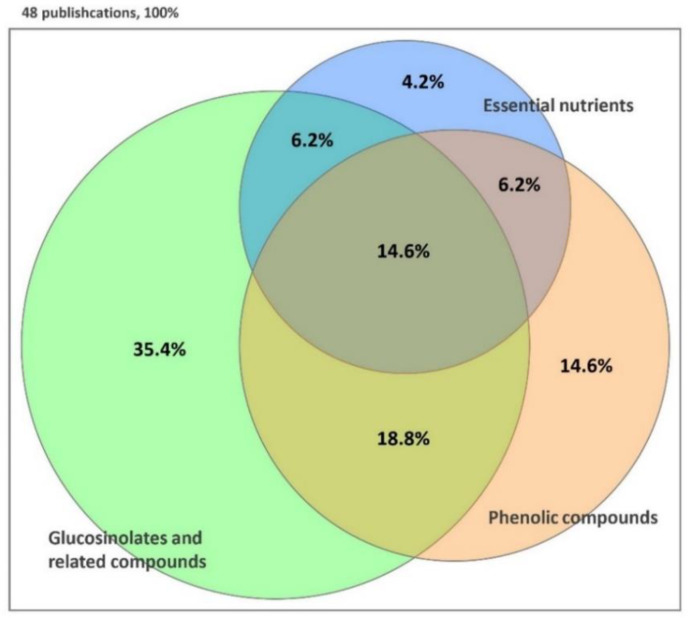
Summary of the bioactive compounds analyzed in the last ten years of broccoli sprouts and microgreens.

**Figure 3 plants-09-00946-f003:**
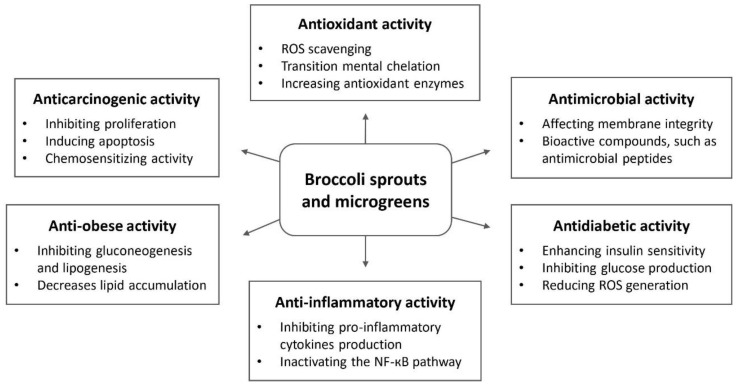
Summary of the biological activities of broccoli sprouts and microgreens with possible mechanisms of action.

**Table 1 plants-09-00946-t001:** Analysis of glucosinolates in broccoli sprouts and microgreens.

No.	Compounds	Molecular Formula	Germination Time	Characterization Method	Ref.
**Aliphatic Glucosinolates**					
1	Sinigrin	C_10_H_17_NO_9_S_2_	4–12 days	HPLC-UV, HPLC-DAD, HPLC-DAD-MS	[14,45,46,47]
2	Gluconapin	C_11_H_19_NO_9_S_2_	3–12 days	HPLC-UV, HPLC-DAD-MS	[14,45,46,48]
3	Progoitrin	C_11_H_19_NO_10_S_2_	3–12 days	HPLC-UV, HPLC-DAD, HPLC-DAD-MS, HPLC-ESI-MS	[14,21,45,46,47,49,50,51]
4	Glucocochlearin	C_11_H_21_NO_9_S_2_	NS	HPLC-ESI-MS	[47]
5	Glucoconringianin	C_11_H_21_NO_9_S_2_	NS	HPLC-ESI-MS	[47]
6	Glucoiberverin	C_11_H_21_NO_9_S_3_	4–12 days	HPLC-DAD	[14,47]
7	Glucosativin	C_11_H_21_NO_9_S_3_	NS	HPLC-ESI-MS	[47]
8	Glucoiberin	C_11_H_21_NO_10_S_3_	3–14 days	HPLC-UV, HPLC-DAD, HPLC-DAD-MS, HPLC-ESI-MS, HPLC-MS/MS	[14,19,21,41,45,46,47,49,50,51,52,53,54,55]
9	Glucoraphenin	C_12_H_21_NO_10_S_3_	8 days	HPLC-DAD	[19]
10	Glucojiaputin	C_12_H_23_NO_9_S_2_	NS	HPLC-ESI-MS	[47]
11	3-Methylbutyl-GLS	C_12_H_23_NO_9_S_2_	NS	HPLC-ESI-MS	[47]
12	3-Methylpentyl-GLS	C_13_H_25_NO_9_S_2_	NS	HPLC-ESI-MS	[47]
13	4-Methylpentyl-GLS	C_13_H_25_NO_9_S_2_	NS	HPLC-ESI-MS	[47]
14	Glucoerucin	C_12_H_23_NO_9_S_3_	3–14 days	HPLC-UV, HPLC-DAD, HPLC-DAD-MS, HPLC-ESI-MS, HPLC-MS/MS, UHPLC-MS/MS	[14,19,21,41,45,46,48,49,51,52,53,54,55,56,57,58,59]
15	Glucoraphanin	C_12_H_23_NO_10_S_3_	3–14 days	HPLC-UV, HPLC-DAD, HPLC-DAD-MS, HPLC-ESI-MS, HPLC-MS/MS, UHPLC-MS/MS	[14,19,21,41,42,45,46,47,48,49,50,51,52,53,54,55,56,57,58,59,60,61,62,63]
16	*n*-Pentyl-GLS	C_13_H_25_NO_9_S_3_	4–12 days	HPLC-DAD	[14]
17	Glucoalyssin	C_13_H_25_NO_10_S_3_	3–12 days	HPLC-DAD, HPLC-MS/MS, HPLC-ESI-MS	[14,47,50,52,57,58]
18	Glucohirsutin	C_16_H_31_NO_10_S_3_	12 days	HPLC-UV	[45]
19	Diglucothiobeinin	C_17_H_31_NO_14_S_4_	NS	HPLC-ESI-MS	[47]
**Aromatic glucosinolates**					
20	Glucosinalbin	C_14_H_19_NO_10_S_2_	4–12 days	HPLC-DAD	[14]
21	Gluconasturtiin	C_15_H_21_NO_9_S_2_	4–12 days	HPLC-DAD	[14]
**Indolic glucosinolates**					
22	*n*-Hexyl-GLS	C_13_H_25_NO_9_S_2_	4–12 days	HPLC-DAD	[14]
23	Glucobrassicin	C_16_H_20_N_2_O_9_S_2_	3–14 days	HPLC-UV, HPLC-DAD, HPLC-DAD-MS, HPLC-ESI-MS, HPLC-MS/MS	[19,21,41,45,46,48,49,50,51,52,54,55,57,58,59]
24	4-Hydroxy-GLB	C_16_H_20_N_2_O_10_S_2_	3–14 days	HPLC-UV, HPLC-DAD, HPLC-DAD-MS, HPLC-ESI-MS, HPLC-MS/MS	[14,19,21,41,46,48,49,50,51,52,54,55,57,58,59]
25	4-Methoxy-GLB	C_17_H_22_N_2_O_10_S_2_	3–12 days	HPLC-UV, HPLC-DAD, HPLC-DAD-MS, HPLC-ESI-MS, HPLC-MS/MS	[14,19,21,41,45,48,49,51,52,55,57,58,59]
26	Neoglucobrassicin	C_17_H_22_N_2_O_10_S_2_	3–14 days	HPLC-UV, HPLC-DAD, HPLC-DAD-MS, HPLC-ESI-MS, HPLC-MS/MS	[14,19,21,41,45,46,48,50,51,52,54,55,57,58,59]
	Total glucosinolates		3–12 days	HPLC-UV, HPLC-DAD, HPLC-DAD-MS, GOD/PAP kit	[14,19,43,44,55,63,64,65]

No., number; Ref., reference; NS, not specified; GLS, glucosinolate; GLB, glucobrassicin; HPLC-UV, high-performance liquid chromatography with an ultraviolet detector; HPLC-DAD, HPLC with a diode array detector; HPLC-DAD-MS, HPLC-DAD coupled to mass spectrometry; HPLC-MS/MS, HPLC coupled to tandem MS; UHPLC-MS/MS, ultra-HPLC coupled to MS/MS; HPLC-ESI-MS, HPLC coupled to electrospray ionization MS.

**Table 2 plants-09-00946-t002:** Analysis of isothiocyanates in broccoli sprouts and microgreens.

No.	Compounds	Molecular Formula	Germination Time	Characterization Method	Ref.
1	Butyronitrile	C_4_H_7_N	3–7 days	GC–MS	[67]
2	Allyl isothiocyanate	C_4_H_5_NS	4 days	GC–MS	[66]
3	2-Methyl-2-nitropropane	C_4_H_9_NO_2_	3–7 days	GC–MS	[67]
4	4-(Methylthio)-butanenitrile	C_5_H_9_NS	5 days	GC–MS	[42]
5	Butyl isothiocyanate	C_5_H_9_NS	4–9 days	GC–MS	[42,44,66,67]
6	Isobutyl isothiocyanate	C_5_H_9_NS	4 days	GC–MS	[19]
7	Iberin	C_5_H_9_NOS_2_	4–8 days	HPLC-DAD, UHPLC-MS/MS, GC–MS	[19,53,66]
8	4-Isothiocyanato-1-butene	C_6_H_9_NS_2_	4–9 days	GC–MS	[42,44,66]
9	3-Methylbutyl isothiocyanate	C_6_H_11_NS	4 days	GC–MS	[66]
10	Isoamyl methyl sulfoxide	C_6_H_14_OS	3–7 days	GC–MS	[67]
11	Erucin	C_6_H_11_NS_2_	3–9 days	UHPLC-MS/MS, GC–MS	[19,44,56,61,67]
12	Sulforaphene	C_6_H_9_NOS_2_	8 days	UHPLC-MS/MS	[19]
13	Sulforaphane	C_6_H_11_NOS_2_	2–12 days	HPLC-UV, HPLC-DAD, HPLC-DAD-MS, HPLC-MS/MS, UHPLC-MS/MS, GC–MS, GC−FID	[19,22,42,44,46,48,53,56,57,59,60,61,62,63,64,66,67,68,69,70,71,72]
14	Indole-3-carbinol	C_9_H_9_NO	8 days	UHPLC-MS/MS	[19]
15	Indole-3-carboxylic acid	C_9_H_7_NO_2_	NS	HPLC-DAD-MS	[46]
16	Indole-3-acetic acid	C_10_H_9_NO_2_	NS	HPLC-DAD-MS	[46]
17	1-Methoxyindole-3-carbaldehyde	C_10_H_9_NO_2_	NS	HPLC-DAD-MS	[46]
	Total isothiocyanates		8 days	UV/Vis, UHPLC-MS	[19,20,73]

UV/Vis, ultraviolet–visible spectrophotometry; GC–MS, gas chromatography combined with mass spectrometry; GC−FID, GC with a flame ionization detector.

**Table 3 plants-09-00946-t003:** Analysis of phenolic compounds in broccoli sprouts and microgreens.

No.	Compounds	Molecular Formula	Germination Time	Characterization Method	Ref.
**Phenolic acids and derivatives**					
1	Benzoic acid	C_7_H_6_O_2_	6 days	HPLC-UV	[76]
2	Salicylic acid	C_7_H_6_O_3_	6 days	HPLC-UV	[76]
3	*p*-Hydroxybenzoic acid	C_7_H_6_O_3_	7–8 days	HPLC-DAD, HPLC-ESI-MS	[21,49]
4	Protocatechuic acid	C_7_H_6_O_4_	5 days	HPLC-UV, HPLC-DAD-MS	[46,78]
5	Gentisic acid	C_7_H_6_O_4_	4 days	HPLC-UV	[70,72]
6	Gallic acid	C_7_H_6_O_5_	5–8 days	HPLC-UV, HPLC-DAD-MS, HPLC-ESI-MS	[21,24,46,49,76,78,79,80]
7	Vanillic acid	C_8_H_8_O	5 days	HPLC-UV	[78]
8	*p*-Coumaric acid	C_9_H_8_O_3_	4–7 days	HPLC-DAD, HPLC-UV	[70,72,78,79,80]
9	Esculetin	C_9_H_6_O_4_	5 days	HPLC-UV	[24]
10	Caffeic acid	C_9_H_8_O_4_	4–7 days	HPLC-UV, HPLC-DAD, HPLC-DAD-MS	[24,43,72,78,79,80]
11	Ferulic acid	C_10_H_10_O_4_	3–14 days	HPLC-UV, HPLC-DAD, HPLC-DAD-MS	[24,54,70,72,76,78,79]
12	Sinapic acid	C_11_H_12_O_5_	3–12 days	HPLC-UV, HPLC-DAD-MS, HPLC-ESI-MS	[14,19,21,43,46,49,54,70,72,76,78,79]
13	Gallic acid hexoside	C_13_H_16_O_10_	7–8 days	HPLC-DAD, HPLC-ESI-MS	[21,49]
14	Gallic acid 4-*O*-glucoside	C_13_H_16_O_10_	7–8 days	HPLC-DAD, HPLC-ESI-MS	[21,49]
15	Sinapoyl malate	C_15_H_16_O_9_	7–8 days	HPLC-DAD, HPLC-ESI-MS	[21,49]
16	Isochlorogenic acid	C_16_H_18_O_9_	4 days	HPLC-UV	[70]
17	Chlorogenic acid	C_16_H_18_O_9_	3–12 days	HPLC-DAD, HPLC-UV	[14,70,72,76,78,79,80]
18	Caffeoyl-quinic acid	C_16_H_18_O_9_	3–8 days	HPLC-DAD, HPLC-DAD-MS, HPLC-ESI-MS	[21,43,49]
19	1-*O*-sinapoyl-β-_D_-glucose	C_17_H_22_O_10_	7–8 days	HPLC-DAD, HPLC-ESI-MS	[21,49]
20	5-*O*-Sinapoylquinic acid	C_18_H_22_O_10_	7–8 days	HPLC-DAD, HPLC-ESI-MS	[21,49]
21	Digalloyl hexoside	C_20_H_20_O_14_	7–8 days	HPLC-DAD, HPLC-ESI-MS	[21,49]
22	1,2-Diferuloylgentiobiose	C_32_H_38_O_19_	7–8 days	HPLC-DAD, HPLC-ESI-MS	[21,49]
23	2-Feruloyl-1-sinapoylgentiobiose	C_33_H_40_O_18_	7–8 days	HPLC-DAD, HPLC-ESI-MS	[21,49]
24	1,2-Disinapoylgentiobiose	C_34_H_42_O_19_	7–8 days	HPLC-DAD, HPLC-DAD-MS, HPLC-ESI-MS	[46,49]
25	1-Sinapoyl-2,2′-diferuloylgentiobiose	C_43_H_46_O_24_	7–8 days	HPLC-DAD, HPLC-ESI-MS	[21,49]
26	2-Feruloyl-1,2’-disinapoylgentiobiose	C_44_H_50_O_22_	7–8 days	HPLC-DAD, HPLC-DAD-MS, HPLC-ESI-MS	[21,46,49]
27	1,2-Disinapoyl-1′-ferulolylgentiobiose	C_44_H_50_O_23_	7–8 days	HPLC-DAD, HPLC-ESI-MS	[21,49]
28	1,2,2′-Trisinapoylgentiobiose	C_45_H_52_O_23_	7–8 days	HPLC-DAD, HPLC-DAD-MS, HPLC-ESI-MS	[21,46,49]
29	Gallotannic acid	C_76_H_52_O_46_	7–8 days	HPLC-DAD, HPLC-ESI-MS	[21,49]
**Flavonoids and derivatives**					
30	Apigenin	C_15_H_10_O_5_	5 days	HPLC-UV	[72,78,80]
31	Kaempferol	C_15_H_10_O_6_	5–12 days	HPLC-UV, HPLC-DAD-MS, HPLC-ESI-MS	[14,46,50,76,78,79,80]
32	Luteolin	C_15_H_10_O_6_	5 days	HPLC-UV	[72,78]
33	Quercetin	C_15_H_10_O_7_	5–12 days	HPLC-UV, HPLC-DAD, HPLC-ESI-MS	[14,24,46,50,72,76,78,79]
34	Myricetin	C_15_H_10_O_8_	5 days	HPLC-UV	[24,72]
35	Astragalin	C_21_H_20_O_11_	7–8 days	HPLC-DAD, HPLC-ESI-MS	[21,49]
36	Rutin	C_27_H_30_O_16_	NS	HPLC-DAD-MS	[46]
37	Robinin	C_33_H_40_O_19_	4 days	HPLC-UV	[70,72]
	Total phenolic compounds		3–14 days	UV/Vis, HPLC-DAD, HPLC-DAD-MS, HPLC-ESI-MS	[20,24,48,51,54,57,64,65,70,73,77,78,80,81,82,83]
	Total flavonoid compounds		3–14 days	UV/Vis, HPLC-DAD, HPLC-DAD-MS, HPLC-ESI-MS	[14,22,24,50,54,64,65,70,77,78,81]

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
