# Peer review of "Bioactive Compounds and Bioactivities of Brassica oleracea L. var. Italica Sprouts and Microgreens: An Updated Overview from a Nutraceutical Perspective"

_plants, 2020, doi:10.3390/plants9080946_

Round 1

Reviewer 1 Report

The article under appreciation is an interesting contribution in the field of plants as it deals with the bioactive compounds and bioactivities of Brassica oleracea L. var. Italica sprouts and microgreens. The study is well performed, but the manuscript needs further elaboration. Some examples are provided below, but there are also many other examples since this field is growing fast.

  1. Introduction lacks a bit of state of the art applied in the field concerning the current status quo in the food industry {Foods, 2020, 9, 523}.
  2. Introduction lacks of state of the art applied in the field { Trends in Food Science & Technology, 2012, 26, 68-87; Trends in Food Science & Technology, 2015, 42, 44-63}.
  3. More feedback concerning emerging technologies {e.g., Food Engineering Reviews, 2015, 7, 357-381; Journal of Food Engineering, 2015, 167, 38-44; Food and Bioproducts Processing 2013, 91, 575-579} could be added in the introduction.
  4. Discussion could include more comparisons with the results of relevant studies {eg Food Chemistry 2018, 254, 150-157; Industrial Crops & Products, 2018, 111, 30-37; Food Chemistry 2019, 296, 47-55; Trends in Food Science & Technology, 2018; 79; 98-105; Trends in Food Science & Technology, 2018; 79; 98-105; Industrial Crops and Products, 2020, 145, 111978}.

Reviewer 2 Report

In my opinion, the topic is interesting and the manuscript has been written with clarity. The organization is good too. Although some parts of the review may be too general, altogether the coverage of the topic is good, such that the readership can obtain a good overview of the importance of the products in health promotion.

.

My major criticism is that that there is no mention at all about carotenoids, despite their importance in agro-food and health and their abundance in chlorophyll-containing tissues such as microgreens. In my view these compounds deserve coverage for a more exhaustive and meaningful review on the topic. There are recent reviews about the importance of these compounds and their derivatives including apocarotenoids in agro-food and health. There are also comprehensive databases about the levels of carotenoids in diverse foods including broccoli. Definitely, carotenoids must be included in a review with a scope like this.

Definitions of terms including essential nutrients, bioactives, functional foods, nutraceuticals and superfoods as well as appropriate references need to be provided in the text.

L37-38. Please, elaborate on the advantages from the sustainability point of view.

Please, be careful when talking about the health benefits of some bioactive compounds, they are usually considered to reduce the risk of developing diseases rather than prevent or having pharmacological actions.

The inclusion of subsections for each group of bioactives would facilitate the organization and reading of the manuscript.

 L181-184 . The usefulness of the in vitro methods used to assess antioxidant activity and the meaning of the results obtained with such approach should be explained. For instance. Are the oxidizing species used produced in the human body? Are all the compounds found in the tested extracts bioavailable? Are the results extrapolable to in vivo conditions? What are the data obtained useful for? This comment is also valid for other activities addressed (anticarcinogenic, antimicrobial, etc.). 

Also, In relation to the potential biological activities in humans, the authors must elaborate on the major  compounds that are actually bioavailable and if they suffer structural modifications before they reach the target locations in the body. Do they all reach the same locations in the cells/other target location?

Reviewer 3 Report

this review works on the seedlings of broccoli (Brassica oleraceaL. var. Italica) which is been recognized by their substantial amount of bioactive constituents, including glucosinolates, phenolic compounds, vitamins, and essential minerals, associated with potential health benefits. this review revised several in vitro and in vivo studies which demonstrated that broccoli seedlings possessed biological properties, including antioxidant, anticancer, antimicrobial, anticancer, anti-inflammatory, anti-obesity and antidiabetic activities. commnets revise exhaustively english language mispelling. please add times span of revision 2007 to 2020?

Round 2

Reviewer 1 Report

Authors followed previous recommendations and the manuscript has been improved.

Reviewer 2 Report

No more comments.